# Development of Accurate Long-lead COVID-19 Forecast

**Wan Yang** [1,2]*, **Jeffrey Shaman**[3,4]

**1** Department of Epidemiology, Mailman School of Public Health, Columbia University, New York, New York, United States of America, **2** Herbert Irving Comprehensive Cancer Center, Columbia University Medical Center, New York, New York, United States of America, **3** Department of Environmental Health Sciences, Mailman School of Public Health, Columbia University, New York, New York, United States of America, **4** Columbia Climate School, Columbia University, New York, New York, United States of America

* wy2202@cumc.columbia.edu

## Abstract

Coronavirus disease 2019 (COVID-19) will likely remain a major public health burden; accurate forecast of COVID-19 epidemic outcomes several months into the future is needed to support more proactive planning. Here, we propose strategies to address three major forecast challenges, i.e., error growth, the emergence of new variants, and infection seasonality. Using these strategies in combination we generate retrospective predictions of COVID-19 cases and deaths 6 months in the future for 10 representative US states. Tallied over >25,000 retrospective predictions through September 2022, the forecast approach using all three strategies consistently outperformed a baseline forecast approach without these strategies across different variant waves and locations, for all forecast targets. Overall, probabilistic forecast accuracy improved by 64% and 38% and point prediction accuracy by 133% and 87% for cases and deaths, respectively. Real-time 6-month lead predictions made in early October 2022 suggested large attack rates in most states but a lower burden of deaths than previous waves during October 2022 –March 2023; these predictions are in general accurate compared to reported data. The superior skill of the forecast methods developed here demonstrate means for generating more accurate long-lead forecast of COVID-19 and possibly other infectious diseases.

## Author summary

Infectious disease forecast aims to reliably predict the most likely future outcomes during an epidemic. To date, reliable COVID-19 forecast remains elusive and is needed to support more proactive planning. Here, we pinpoint the major challenges facing COVID-19 forecast and propose three strategies. Comprehensive testing based on retrospective forecasts shows the forecast approach using all three strategies consistently outperforms a baseline approach without these strategies across different variant waves and locations in the United States for all forecast targets, improving the probabilistic forecast accuracy by ~50% and point prediction accuracy by ~100%. The superior skills of the forecast methods

**Data Availability Statement:** Data and model code are publicly available at https://github.com/wan-yang/covid_long_lead_forecast.

**Funding:** This study was supported by the National Institute of Allergy and Infectious Diseases (AI145883 to WY and JS; AI163023 to JS), the

Centers for Disease Control and Prevention (CDC) and the Council of State and Territorial Epidemiologists (CSTE; contract no.: NU38OT00297 to WY and JS), and the CDC Center for Forecasting and Outbreak Analytics (contract no.: 75D30122C14289 to JS and WY). The funders had no role in study design, data collection and analysis, decision to publish, or preparation of the manuscript.

**Competing interests:** I have read the journal's policy and the authors of this manuscript have the following competing interests: JS and Columbia University disclose partial ownership of SK Analytics; JS discloses consulting for BNI.

developed here demonstrate means for generating more accurate long-lead COVID-19 forecasts. The methods may be also applicable to other infectious diseases.

## Introduction

The severe acute respiratory syndrome coronavirus 2 (SARS-CoV-2) emerged in late 2019, causing the coronavirus disease 2019 (COVID-19) pandemic. Since its onset, mathematical modeling has been widely applied to generate projections of potential pandemic trajectories, including for cases, hospitalizations, and deaths. These model projections are often based on specific assumptions (i.e., scenarios) regarding critical factors affecting transmission dynamics. For example, the scenarios often include a combination of public health policies [e.g., non-pharmaceutical interventions (NPIs) including lockdown/reopening and masking, and vaccinations], population behavior (e.g., adherence to the policies and voluntary preventive measures), and anticipated changes in the epidemiological properties of SARS-CoV-2 variants [1–4]. While such efforts have provided overviews of the potential outcomes under various scenarios, they do not assign likelihoods to the scenarios/projected trajectories, and the most likely trajectory is typically not known until the outcome is observed. That is, scenario projection is not equivalent to calibrated infectious disease *forecast*, which aims to reliably predict the most likely future outcomes during an epidemic. As COVID-19 will likely remain a major public health burden in the years to come, sensible forecast of the health outcomes several months in the future is needed to support more proactive planning.

Compared to forecast of epidemic infections (e.g., influenza), a number of additional challenges exist for long-lead COVID-19 forecast. First, SARS-CoV-2 new variants will likely continue to emerge and remain a major source of uncertainty when generating COVID-19 forecasts [5,6]. As has been observed for the major variants of concern (VOCs) reported to date (i.e., Alpha, Beta, Gamma, Delta, and Omicron), future new variants could arise at any time, could quickly displace other circulating variants, and could be more contagious than pre-existing variants, and/or erode prior infection- and/or vaccination-induced immunity to affect underlying population susceptibility. Further, multiple new variants could arise successively to cause multiple waves during a time span of, e.g., 6 months. Such frequent emergence and fast turnover of circulating variants is in stark contrast with epidemic infections. Second, many infections for other respiratory viruses tend to occur during a certain season of the year and as such, the seasonality can be incorporated to improve forecast accuracy [7,8] as well as restrict the forecast window to the epidemic season (e.g., influenza during the winter in temperate regions). Potential seasonality for SARS-CoV-2 is still not well characterized. For instance, in the US, where larger waves have occurred in the winter during 2020–2022, smaller summer waves have also occurred (e.g., the initial Delta wave and Omicron subvariant waves; Fig 1). Third, given the unknown timing of new variant emergence, year-round, long-lead COVID-19 forecast will likely be needed. These unknowns also necessitate wider parameter ranges using a forecast ensemble to account for the uncertainty, which over long forecast horizons could lead to greater error growth and poorer predictive accuracy.

In this study, we aim to address the above challenges and develop sensible approaches that support long-lead prediction of COVID-19 epidemic outcomes. We propose three strategies for improving forecast accuracy and test the methods in combination by generating retrospective forecasts of COVID-19 cases and deaths in 10 representative states in the US (i.e., California, Florida, Iowa, Massachusetts, Michigan, New York, Pennsylvania, Texas, Washington, and Wyoming; Fig 1). Relative to a baseline approach, the forecast approach based on our

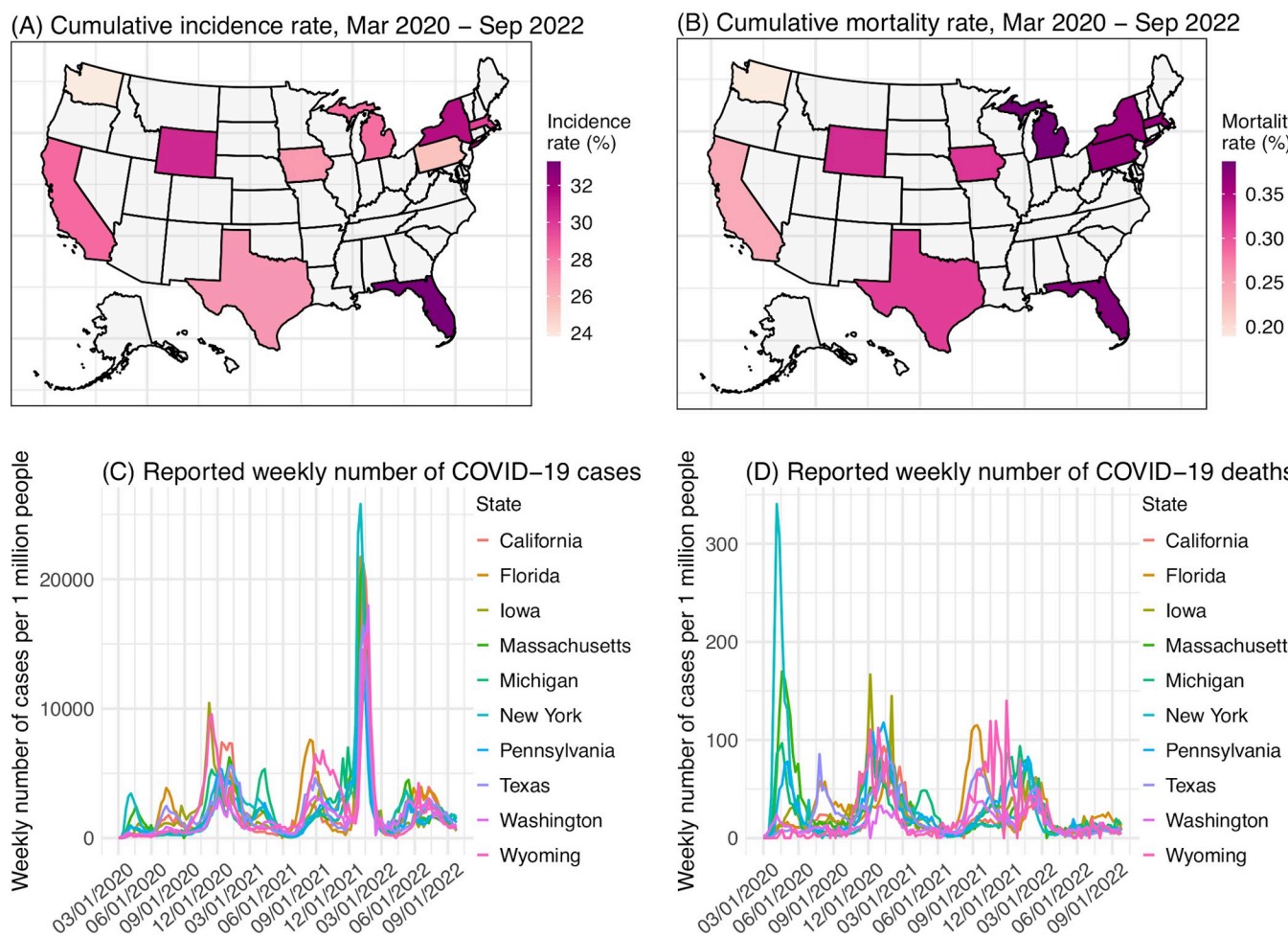

**Fig 1. Geospatial distribution of the 10 states and overall COVID-19 outcomes.** Heatmaps show reported cumulative COVID-19 incidence rates (A) and COVID-19-associated mortality rates (B) in the 10 states included in this study. Line plots show reported weekly number of COVID-19 cases (C) and COVID-19-associated deaths (D) during the study period, for each state. The maps are generated using the "usmap" R package with data from the US Census (https://www.census.gov/geographies/mapping-files/time-series/geo/cartographic-boundary.html).

strategies largely improves forecast accuracy (64%/38% higher probabilistic log score for cases/deaths, and 133%/87% higher point prediction accuracy for cases/deaths, tallied over 25,183 evaluations of forecasts initiated during July 2020 –September 2022, i.e., from the end of the initial wave to the time of this study). These results highlight strategies for developing and operationalizing long-lead COVID-19 forecasts with greater demonstrated accuracy and reliability. In addition, we generate real-time COVID-19 forecasts for October 2022 –March 2023 (roughly covering the 2022–2023 respiratory virus season) and evaluate these forecasts using data reported 6 months later.

## Results

### Proposed strategies for long-lead COVID-19 forecast

To address the three challenges noted above, we propose three strategies in combination. Details are provided in Methods. Here, we summarize the idea behind each strategy. The first strategy is to constrain error growth during the extended forecast period. As noted above, the multiple uncertainties regarding SARS-CoV-2 (e.g., new variant properties) necessitate wider

distributions of the state variables (e.g., population susceptibility) and parameters (e.g., virus transmissibility) at the point of forecast initiation. With a wider forecast ensemble, some ensemble members could predict earlier, large outbreaks, which if premature, would deplete modeled susceptibility and incorrectly preclude outbreaks later on. More generally, like other infections, COVID-19 epidemics often grow exponentially at first, triggering exponential changes in susceptibility and other state variables, which in turn feed back on the longer epidemic trajectory. Such infectious disease dynamics imply forecast error can also grow exponentially, leading to accelerated degradation of forecast accuracy after the first few weeks. To counteract this exponential error growth, we propose to apply a multiplicative factor $\gamma < 1$ to the covariance of the forecast state variables (e.g., susceptibility) while retaining the ensemble mean (see Methods). In so doing, we can represent initial uncertainty with a broad forecast ensemble while countering excessive error growth of the state variables. This strategy is similar to covariance inflation during system optimization [9,10], where a $\gamma > 1$ is applied to the covariance to counteract over-narrowing of the model ensemble. Since here it acts in the opposite direction, we refer to this technique as deflation. Fig 2A shows example forecasts with deflation compared to without it.

The second strategy is to anticipate the impact of new variants. Genomic sequencing data can support prediction of the impact of a new variant a few weeks in the future (see Methods and S1 Text); however, variant displacement and competition dynamics can occur unexpectedly beyond those first few weeks, rendering historical data less relevant. Thus, for weeks farther in the future, instead of predicting specific new variants, we propose to use a set of heuristic rules to anticipate their likely emergence timing and impact on population susceptibility and virus transmissibility. Specifically, for the timing, we reason that new variants are more likely to emerge/circulate 1) after a large local wave when more infections could lead to more mutations, which could be timed based on local outbreak intensity during the preceding months; and 2) during a time when a large part of the world is experiencing a large wave, which also could lead to new mutations. For instance, in the US, this could be during the winter when local large waves tend to occur, or the summer when places in the southern hemisphere are amid their winter waves. As such, this could be timed based on the calendar. For the new variant impact, we observe that, new variant circulation often results in gradual increases in susceptibility (e.g., a few percentages per week, based on estimates from New York City during VOC waves), possibly due to the substantial population immunity accumulated via infections and vaccinations. Similarly, changes in transmissibility also tend to occur gradually. As such, we propose to apply small increases to the model population susceptibility and transmissibility during those two plausible times. The rationale here is to anticipate the more common, non-major changes so as not to overpredict, because major VOCs causing dramatic changes are rarer and difficult to predict. Fig 2B shows example forecasts with these new variant settings compared to without them.

The third strategy accounts for seasonality. Instead of assuming a specific epidemic timing, we model the seasonal risk of SARS-CoV-2 infection based on plausible underlying drivers of infection seasonality for common respiratory viruses. Specifically, studies have shown that respiratory viruses including SARS-CoV-2 are sensitive to ambient humidity and temperature conditions, which could in turn modulate their survival and transmission [11–15]. Accordingly, we developed a climate-forced model that includes both humidity and temperature to capture the reported virus response and parameterized the model based on long-term epidemic data observed for influenza [16]. When modified and applied to account for SARS-CoV-2 seasonality in a model-inference framework, the estimated seasonal trends are able to capture the effects of different climate conditions (e.g., for UK, Brazil, India, and South Africa [17–19]). Here, we thus propose to apply this model and local climate data to estimate

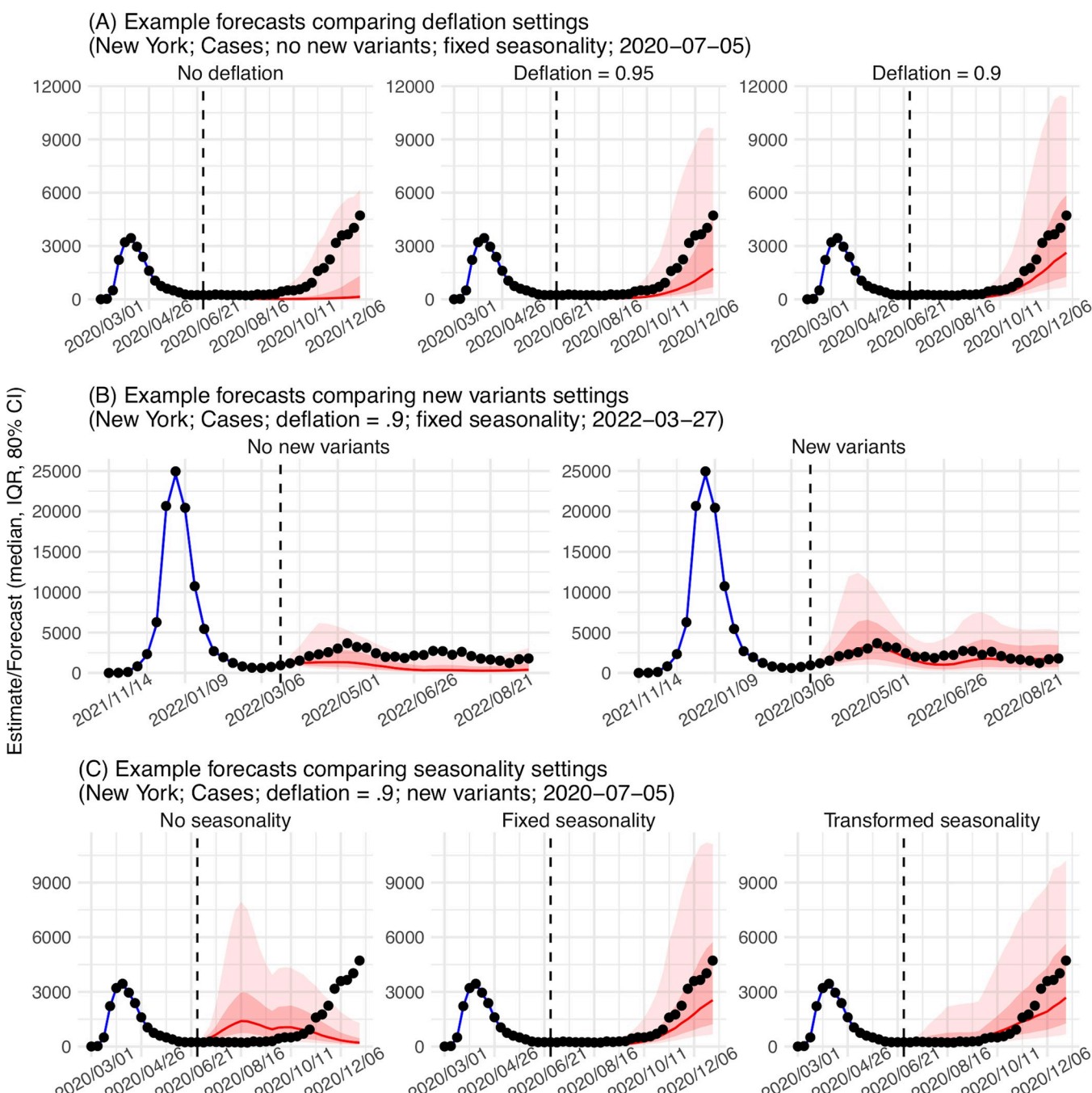

**Fig 2. Example forecasts.** Vertical dashed lines indicate the week of forecast. Dots show reported weekly cases per 1 million people; only those to the left of the vertical lines are used to calibrate the model and those to the right of the vertical lines are plotted for comparison. Blue lines and blue areas (line = median; darker blue = 50% CI; lighter blue = 80% CI) show model training estimates. Red lines and red areas (line = median; dark red = 50% CI; lighter red = 80% CI) show model forecasts using model settings as labeled in the subtitles.

SARS-CoV-2 seasonality in each location (referred to as "fixed seasonality"; see estimates for the 10 states in S1 Fig). In addition, as the model is parameterized based on influenza data, this estimated seasonality could differ from the true SARS-CoV-2 seasonality. To test this, we propose an alternative seasonality form that transforms the fixed seasonality trend to allow a more

flexible phase timing and structure of seasonality (referred to as "transformed seasonality"; see details in S1 Text and examples for the 10 states in S1 Fig). Fig 2C shows a comparison of the two seasonality forms and example forecasts with no seasonality and the two seasonality forms.

We test the above three strategies in combination, including three deflation settings (i.e., setting $\gamma$ = 1, 0.95, and 0.9), two new variant settings (i.e., assuming no new variants vs. anticipating new variants per the rules noted above), and three seasonality settings (i.e., no, fixed, and transformed seasonality): in total, 12 (= $3 \times 2 \times 3$) forecast approaches. To compare the performance of the 12 forecast approaches, we generated retrospective forecasts for 10 states, from July 2020 –August 2021 (Pre-Omicron period; 65 weeks in total) and December 2021 – September 2022 (Omicron period; ~37 weeks). For each week, a forecast for the following 26 weeks (~6 months) is generated after model training using data up to the week of forecast. We then evaluate the accuracy predicting the weekly number of cases and deaths during each of the 26 weeks (i.e. 1- to 26-week ahead prediction), as well as the peak timing (i.e. the week with the highest cases/deaths), peak intensity, cumulative number of cases and deaths over the 26 weeks.

For the forecast comparisons below, we evaluated probabilistic forecast accuracy using log score, i.e., the logarithm of the probability correctly assigned to the true target (see Methods and S1 Text). We also evaluated point prediction accuracy–assigning value 1 (i.e., accurate) to a forecast if the point prediction is within ±25% of the observed case/death count or within ±1 week of the observed peak week, and 0 (i.e., inaccurate) otherwise; as such, when averaged over all forecasts, this accuracy gives the percentage of forecasts a point prediction is accurate within these tolerances. Thus, both higher log score and higher point prediction accuracy indicate superior forecast performance.

## The impact of deflation

Compared to forecasts with no deflation ($\gamma$ = 1), applying a small deflation ($\gamma$ = 0.95 or 0.9) consistently improves forecast performance (Fig 3, first two panels). This improvement is evident across all locations and combinations of the other two model settings (i.e., new variants and seasonality). The improvement is most pronounced for the long-lead (i.e., 17- to 26-week ahead) weekly forecasts and overall intensity-related targets (i.e., the totals accumulated over 26 forecast weeks and peak intensity; S2 and S3 Figs), indicating deflation is able to effectively reduce error growth accumulated over time. We note that, as deflation works by reducing forecast spread, the forecast ensemble can also become under-dispersed, assign zero probability to the true target (most notably, for peak week), and in turn produce a lower log score (see S2 Fig, the 3$^{rd}$ row of each heatmap for peak week). Given this trade-off, we did not test deflation factors <0.9.

Overall, relative to forecasts without deflation ($\gamma$ = 1), log scores aggregated over all locations and targets were 18–20% higher for forecasts of cases (range of relative change across all combinations of the other two model settings; Table A in S1 Text) and 7–8% higher for forecasts of mortality when a deflation factor $\gamma$ of 0.95 was applied. The log scores further increased when using $\gamma$ = 0.9, to 34–43% higher (than $\gamma$ = 1) for forecasts of cases and 13–17% higher for forecasts of mortality. The improvement of point prediction accuracy was more pronounced. Aggregated over all locations and targets, point prediction accuracy was 33–63% higher for forecasts of cases and 24–40% higher for forecasts of mortality when using a deflation factor $\gamma$ of 0.9, relative to using no deflation (Table A in S1 Text). As such, we use $\gamma$ = 0.9 as the best-performing setting for subsequent analyses.

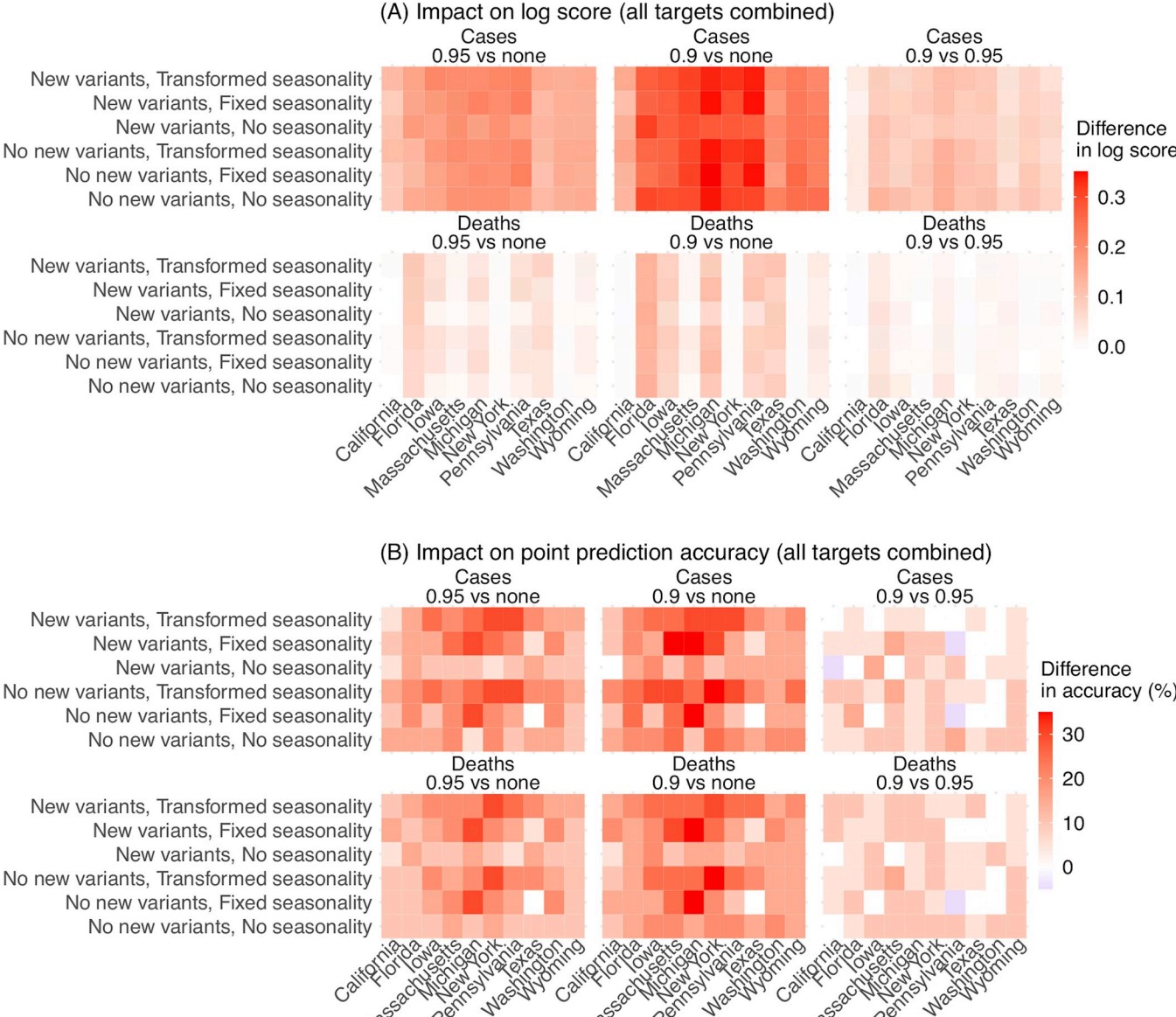

**Fig 3. Impact of deflation on forecast performance.** Heatmaps show the differences in mean log score (A) or point prediction accuracy (B) between all forecast approaches with different deflation settings (deflation factor γ = 0.95 vs none in the 1st column, 0.9 vs none in the 2nd column, and 0.9 vs 0.95 in the 3rd column; see panel subtitles). Results are aggregated for each forecast approach (see specific settings of new variants and seasonality in the y-axis labels) and location (x-axis) over all forecast targets and forecast weeks, for cases (1st row) and deaths (2nd row), separately. For each pairwise comparison (e.g., 0.95 vs none), a positive difference in log score or point prediction accuracy indicates the former approach (e.g., 0.95) outperforms the latter (e.g., none).

## Impact of the new variant settings

To ensure consistency and avoid over-fitting, we applied the same set of heuristic rules on new variant emergence timing and impact, as noted above, throughout the entire study period (i.e., including weeks before the emergence of SARS-CoV-2 VOCs). We expect the new variant settings to improve forecast performance during VOC waves but have a less pronounced or no effect during the 2nd wave (roughly, fall/winter 2020–2021), prior to VOC emergence. Indeed, overall, for the 2nd wave, forecast systems with the new variant settings (referred to as "new variant model") had similar performance as those assuming no new variants (relative changes

in log score and accuracy: -2% to 0.6%, Table B in S1 Text; Fig 4). However, for the VOC waves, applying the new variant settings improved forecast performance during the Alpha wave (roughly, spring 2021), Delta wave (roughly, summer/fall 2021), and Omicron wave (after December 2021; note no further desegregation was made for Omicron subvariant waves due to small sample sizes; Fig 4). For forecasts of cases, the improvement was consistently seen for all three VOC waves (relative changes in log score and accuracy all > 0, Table B in S1 Text). The relative increases of log score were up to 119% during the Delta wave and up to 37% during the Omicron wave; the relative increases of point prediction accuracies were up to 89% during the Delta wave and up to 96% during the Omicron wave (Table B in S1 Text and Fig 4).

For forecasts of mortality, the new variant model had higher log scores during the Alpha and Delta waves, as well as higher point prediction accuracies during all three VOC waves (Table B in S1 Text). However, log scores during the Omicron wave were similar for both

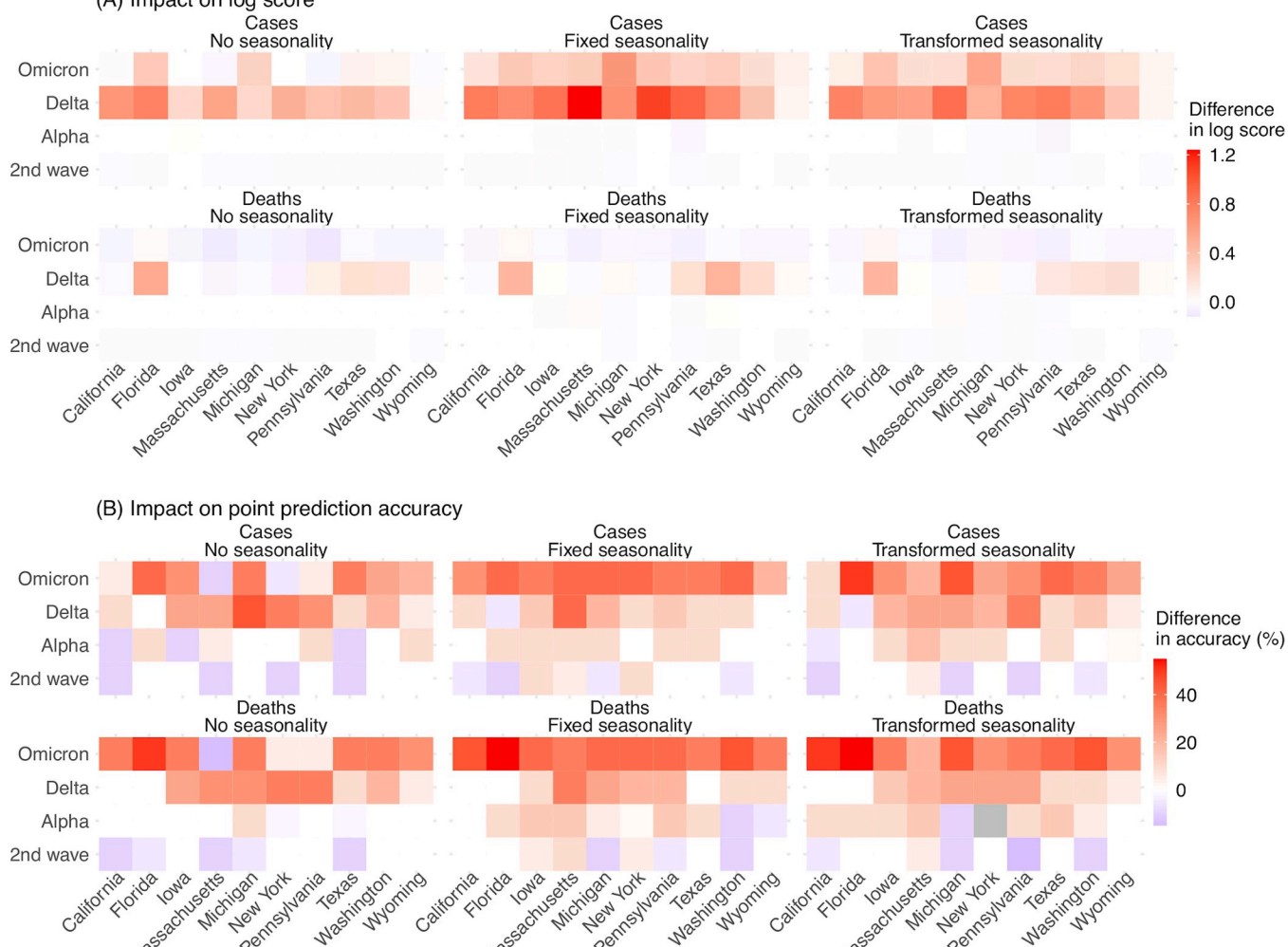

**Fig 4. Impact of new variant settings on forecast performance.** Heatmaps show the differences in mean log score (A) or point prediction accuracy (B) between forecast approaches with vs without anticipation of new variant emergence. All forecasts here were generated using a deflation factor of 0.9. Results are aggregated for each forecast approach (see specific setting of seasonality in panel subtitles), variant wave (y-axis), and location (x-axis) over all forecast targets and forecast weeks for cases (1st row) and deaths (2nd row), separately. A positive difference indicates superior performance of the forecast approach with anticipation of new variant emergence.

models (e.g., overall log scores: -0.22 for the new variant model and -0.17 for the baseline, both assuming no seasonality); in addition, the log scores were slightly lower for the new variant model, likely due to the lower COVID-19-associated deaths during the Omicron wave.

Aggregated over all waves, targets, and locations, the new variant model had 17–34% and 3–8% higher log scores and 23–28% and 22–25% higher point prediction accuracies for forecasts of cases and deaths, respectively.

## Impact of seasonality forms

Based on the above results, we focus on forecasts generated using the new variant model with a deflation factor of 0.9 to examine the three approaches to forecasting the effects of seasonality. As noted above, seasonality aims to capture changes in infection risk in response to environmental conditions (here, ambient humidity and temperature); for common respiratory viruses (e.g., influenza), infection risk in temperate regions is often higher during cold-dry winter months (i.e., respiratory virus season, roughly mid-October to mid-April in the US), and lower during the rest of the year (i.e., off season). Both the fixed and transformed seasonality models consistently improved forecast performance during the respiratory virus season relative to the no seasonality approach across all locations and targets (Fig 5A for log score and 5B for point prediction accuracy, first two panels; Table C in S1 Text). However, if only analyzing the off season, both models had worse performance compared to the model assuming no seasonality (Fig 5, 4th and 5th panels). Segregating the forecasts made for the off season by wave shows that both seasonality models continued to outperform the no seasonality model during summer/fall 2020 (grouped with the 2nd wave); worse performance occurred during summer/fall 2021 (the Alpha and Delta waves) and summer 2022 (Omicron subvariants; Table C in S1 Text). These results suggest that the seasonality models are able to capture the seasonal risk of SARS-CoV-2 infection, and that the degraded performance during the off season may be due to challenges anticipating the initial surge of VOCs occurring during those times.

Tallied over all time periods, in general, the seasonality models outperformed the no seasonality model (Table C in S1 Text for all locations combined and Table D in S1 Text for each location). Comparing the two seasonality forms, the transformed seasonality model outperformed the fixed seasonality model overall (Table C in S1 Text). Compared to the no seasonality model, the transformed seasonality had 14% higher log score for forecasts of mortality, and 26% and 18% higher point prediction accuracies for cases and mortality, respectively (Table C in S1 Text). As noted above, the improvement during the respiratory virus season was more substantial (Tables C and D in S1 Text; Fig 5).

## Combined impact of deflation, new variant settings, and seasonality forms

We now examine the forecast performance using the combined best-performing approach (i.e., applying deflation with $\gamma = 0.9$, the new variant rules, and the transformed seasonality form), compared to the baseline approach (i.e., no deflation, no new variants, and no seasonality). In addition to the large uncertainties surrounding SARS-CoV-2 (e.g., new variants), there are also large spatial heterogeneities. For example, across the 10 states included here, population density ranged from 6 people per square mile in Wyoming to 884 per square mile in Massachusetts (2020 data [20]); climate conditions span temperate (e.g., New York) and subtropical climates (e.g., Florida; S1 Fig). Given the uncertainties and spatial heterogeneities, the robustness of any forecast approach is particularly important.

First, we examine the consistency of forecast performance over different variant periods. During the pre-Omicron period (here, July 2020 –Dec 2021), the combined approach consistently outperformed the baseline approach across all states, for both forecasts of cases and deaths

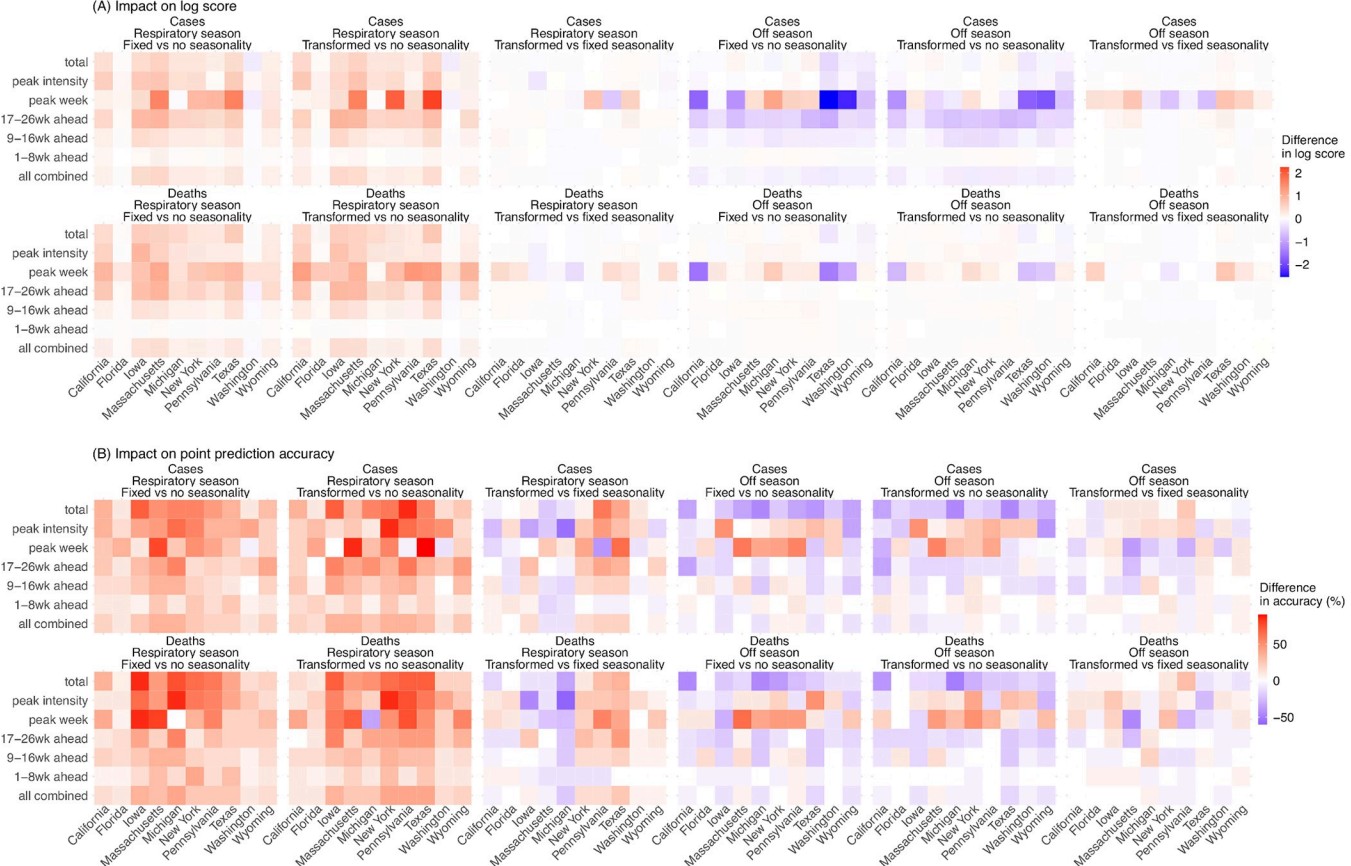

**Fig 5. Impact of seasonality settings on forecast performance.** Heatmaps show the differences in mean log score (A) or point prediction accuracy (B), between pairs of forecast approaches with different seasonality settings (see panel subtitles). All forecasts here were generated using a deflation factor of 0.9 and the new variant setting. Results are aggregated for each forecast target (y-axis) and location (x-axis), over either the respiratory virus season (first 3 columns) or the off season (last 3 columns), for cases (1st row) and deaths (2nd row), separately. For each pairwise comparison (e.g., fixed vs no seasonality), a positive difference in log score or point prediction accuracy indicates the former approach (e.g., with fixed seasonality) outperforms the latter (e.g., with no seasonality).

(Fig 6A); log scores improved by 85% overall for cases (range from 39% in Washington to 134% in Florida) and by 62% overall for deaths (range from 24% in Washington to 117% in Florida; Table 1). During the Omicron period (here, Dec 2021–September 2022), improvements were smaller but consistent across the 10 states for forecast of cases (Fig 6A; note that only 16–20 forecasts of long-lead targets were evaluated here, as observations are incomplete); as noted above, due to the much lower mortality during the Omicron period, both the best-performing and baseline forecasts of mortality had similar log scores (e.g., median difference = -0.02, Table 1). The overall consistency of the performance indicates that the best-performing forecast approach is robust for forecasting long-lead COVID-19 epidemic outcomes for different variants.

Second, we examine the forecast performance during the US respiratory virus season (here, mid-October to mid-April) when larger COVID-19 waves have occurred. Tallied over all weeks during the respiratory virus season, the best-performing approach outperformed the baseline approach for all 10 states (Fig 6B); log scores increased by 105% for cases (range from 32% in Washington to 213% in Massachusetts) and by 85% for mortality (range from 19% in Washington to 158% in Iowa; Table 1). During the off season, the best-performing approach also generally outperformed the baseline approach (Table 1).

Third, we examine the accuracy predicting different epidemic targets. The best-performing forecast approach consistently improved point prediction accuracy for all targets for all 10

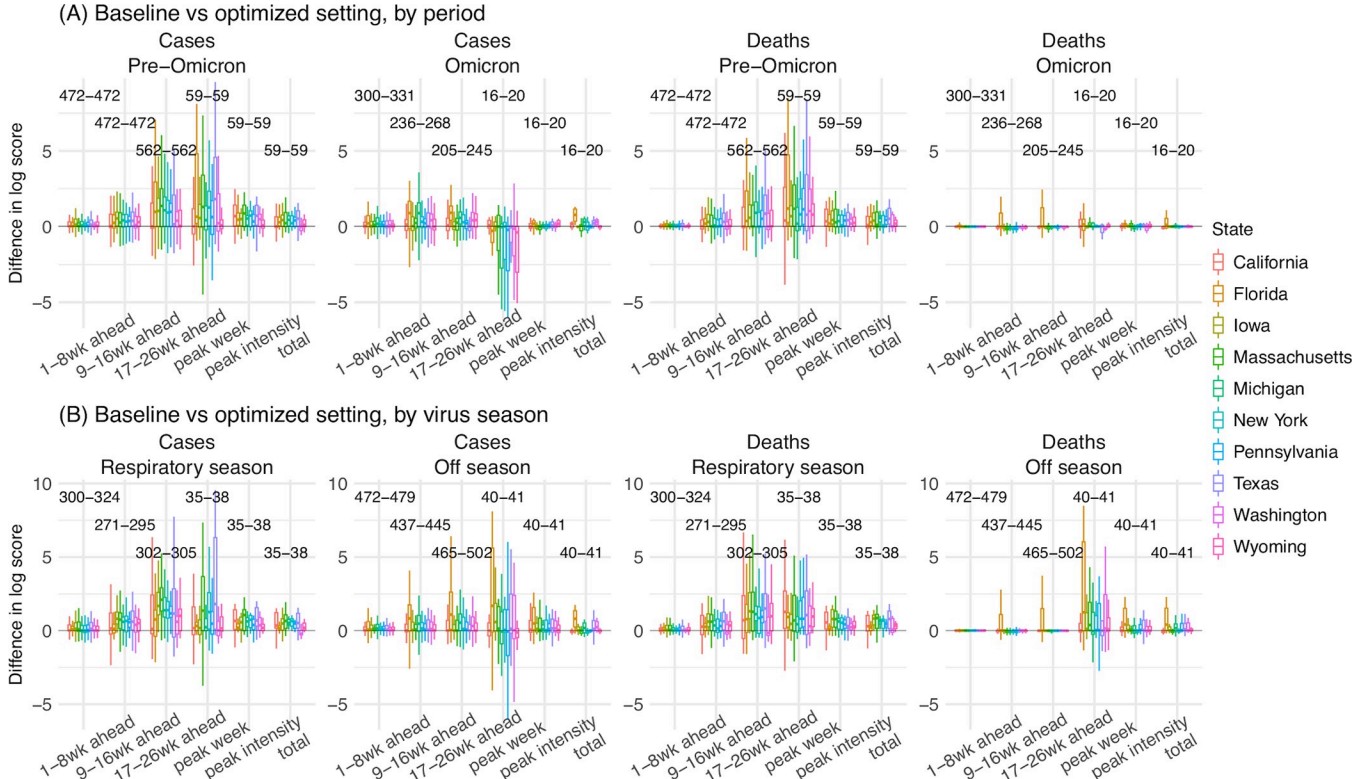

**Fig 6. Probabilistic forecast accuracy of the best-performing and baseline forecast approaches.** Boxplots show the distributions of pair-wise difference in log score by variant period (A) or respiratory virus season (B; see panel subtitles). Results are aggregated by location (color-coded for each state) and forecast target (x-axis), for cases and deaths (see panel subtitles), separately. The numbers show the range of number of evaluations of each forecast target (e.g., 59 predictions of peak week during the pre-Omicron period, for each state; 16–20 predictions of peak week during the Omicron period, depending on the timing of Omicron detection in each state). A positive difference indicates superior log score of the best-performing forecast approach.

states (Fig 7 and Table 2). In addition, the improvement was more substantial for long-lead targets (e.g., 9- to 16-week ahead and 17- to 26-week ahead forecasts, peak week, peak intensity, and the cumulative totals). For instance, the best-performing approach increased accuracy from 24% to 42% (35% to 56%) predicting the peak week of cases (deaths), from 20% to 40% (30% to 50%) predicting the peak intensity of cases (deaths), roughly a 2-fold improvement for these two long-lead targets. The improvements were even more substantial for 17–26 week ahead forecasts and the cumulative totals over the entire 26 weeks (by 3- to 22-fold, Fig 7 and Table 2). We note the forecasts here were generated retrospectively with information that may not be available in real time and thus likely are more accurate as a result. Nonetheless, with the same information provided to both forecast approaches, the comparison here demonstrates the large improvement in forecast accuracy using the best-performing forecast approach.

## Forecast performance compared with ARIMAX models

To benchmark the performance of the forecast approaches developed here, we also generated retrospective forecasts using Auto-Regressive Integrated Moving Average (ARIMA) models. Compared to the best-performing ARIMAX model (identified from 5 models with different settings; see Methods and Table E in S1 Text), our baseline approach (i.e., no deflation, no new variants, and no seasonality) performed similarly well whereas our best-performing approach (i.e., applying deflation with γ = 0.9, the new variant rules, and the transformed seasonality form) had much superior performance (Table F in S1 Text).

**Table 1. Comparison of probabilistic forecast accuracy by the best-performing and the baseline forecast approaches.** Numbers show the relative difference in mean log score computed using Eq 6, the median of pairwise difference in log score (95% CI of the median); asterisk (*) indicates if the median is significantly >0 or <0 at the α = 0.05 level, per a Wilcoxon rank sum test. Positive numbers indicate superior performance of the best-performing forecast approach.

| State | Measure | All | Pre-Omicron period | Omicron period | Respiratory season | Off season |
|---|---|---|---|---|---|---|
| All | Cases | 63.6%, 0.34 (0.33, 0.35) * | 85.3%, 0.43 (0.42, 0.44) | 27.2%, 0.21 (0.2, 0.22) * | 105%, 0.53 (0.51, 0.54) | 40.6%, 0.22 (0.21, 0.23) * |
| All | Deaths | 38%, 0.07 (0.06, 0.08) * | 61.7%, 0.25 (0.24, 0.27) | 0.184%, -0.018 (-0.02, -0.016) * | 84.5%, 0.42 (0.41, 0.44) | 13.7%, 0 (0, 0) * |
| California | Cases | 46.3%, 0.18 (0.16, 0.21) * | 65.8%, 0.26 (0.22, 0.32) | 15.9%, 0.11 (0.08, 0.14) * | 93.7%, 0.35 (0.27, 0.43) | 21.2%, 0.13 (0.1, 0.15) * |
| California | Deaths | 29.9%, 0.01 (0, 0.01) * | 50.6%, 0.06 (0.04, 0.1) * | -1.26%, -0.028 (-0.032, -0.024) * | 91.4%, 0.34 (0.26, 0.42) | 0.225%, -0.0095 (-0.013, -0.006) * |
| Florida | Cases | 119%, 0.51 (0.46, 0.55) * | 134%, 0.42 (0.37, 0.49) | 90.7%, 0.63 (0.57, 0.68) * | 48.9%, 0.23 (0.2, 0.27) | 183%, 0.77 (0.71, 0.83) * |
| Florida | Deaths | 84.5%, 0.23 (0.2, 0.26) * | 117%, 0.28 (0.24, 0.33) | 33.6%, 0.11 (0.06, 0.18) * | 58.5%, 0.27 (0.24, 0.31) | 104%, 0.17 (0.12, 0.26) * |
| Iowa | Cases | 69.8%, 0.4 (0.37, 0.44) * | 108%, 0.59 (0.55, 0.63) | 10.6%, 0.09 (0.06, 0.13) * | 172%, 0.84 (0.77, 0.89) | 24.7%, 0.19 (0.16, 0.22) * |
| Iowa | Deaths | 50.6%, 0.19 (0.16, 0.22) * | 86.4%, 0.43 (0.39, 0.47) | -4.49%, -0.024 (-0.03, -0.018) * | 158%, 0.71 (0.66, 0.77) | 5.74%, 0 (0, 0.01) * |
| Massachusetts | Cases | 84.1%, 0.42 (0.38, 0.46) * | 131%, 0.66 (0.61, 0.72) | 17.6%, 0.17 (0.13, 0.2) * | 213%, 0.91 (0.83, 0.99) | 29%, 0.16 (0.14, 0.19) * |
| Massachusetts | Deaths | 40.1%, 0.04 (0.03, 0.06) * | 68.9%, 0.28 (0.23, 0.34) | -3.32%, -0.031 (-0.041, -0.022) * | 130%, 0.61 (0.55, 0.67) | 0.439%, -0.005 (-0.009, -0.002) * |
| Michigan | Cases | 76.4%, 0.48 (0.45, 0.51) * | 86.9%, 0.49 (0.45, 0.52) | 55.9%, 0.46 (0.4, 0.51) * | 119%, 0.62 (0.57, 0.68) | 53.2%, 0.39 (0.35, 0.42) * |
| Michigan | Deaths | 36.9%, 0.12 (0.09, 0.14) * | 60.9%, 0.32 (0.28, 0.35) | -2.89%, -0.029 (-0.036, -0.022) * | 85.6%, 0.48 (0.44, 0.52) | 12.2%, -0.0044 (-0.0095, -0.00048) * |
| New York | Cases | 60.7%, 0.37 (0.34, 0.4) * | 79.6%, 0.47 (0.44, 0.5) | 28.8%, 0.22 (0.19, 0.25) * | 117%, 0.63 (0.57, 0.69) | 31.1%, 0.21 (0.18, 0.24) * |
| New York | Deaths | 20.6%, 0.01 (0.01, 0.02) * | 35.5%, 0.09 (0.06, 0.15) | -4.31%, -0.025 (-0.032, -0.019) * | 61.1%, 0.37 (0.33, 0.4) | -0.7%, -0.0056 (-0.009, -0.0029) * |
| Pennsylvania | Cases | 49.2%, 0.32 (0.3, 0.35) * | 72.4%, 0.45 (0.42, 0.48) | 12.1%, 0.16 (0.13, 0.18) * | 94.9%, 0.56 (0.52, 0.6) | 24.7%, 0.18 (0.16, 0.2) * |
| Pennsylvania | Deaths | 30.5%, 0.12 (0.09, 0.15) * | 51.7%, 0.3 (0.27, 0.33) | -3.29%, -0.024 (-0.034, -0.016) * | 71.9%, 0.41 (0.38, 0.44) | 8.45%, 0.01 (0, 0.01) * |
| Texas | Cases | 74.3%, 0.32 (0.29, 0.36) * | 120%, 0.51 (0.46, 0.57) | 9.88%, 0.07 (0.04, 0.12) * | 158%, 0.64 (0.56, 0.71) | 34%, 0.18 (0.15, 0.21) * |
| Texas | Deaths | 59.3%, 0.17 (0.14, 0.21) * | 103%, 0.47 (0.42, 0.52) | -1.88%, -0.012 (-0.015, -0.009) * | 141%, 0.67 (0.6, 0.74) * | 20.5%, 0.01 (0.01, 0.02) * |
| Washington | Cases | 38.5%, 0.26 (0.23, 0.28) * | 38.8%, 0.24 (0.21, 0.27) | 37.8%, 0.28 (0.25, 0.32) * | 31.7%, 0.28 (0.24, 0.32) | 43.3%, 0.24 (0.2, 0.26) * |
| Washington | Deaths | 13.8%, 0 (-0.0015, 0) | 24.1%, 0.04 (0.02, 0.08) * | -4.22%, -0.028 (-0.034, -0.022) * | 19.1%, 0.07 (0.03, 0.13) | 10.4%, -0.001 (-0.0029, 0.00093) |
| Wyoming | Cases | 34.5%, 0.21 (0.19, 0.23) * | 45.7%, 0.25 (0.23, 0.28) | 13.4%, 0.11 (0.07, 0.15) * | 73.8%, 0.42 (0.38, 0.46) | 13.7%, 0.09 (0.07, 0.12) * |
| Wyoming | Deaths | 26.8%, 0.05 (0.04, 0.08) * | 42.7%, 0.19 (0.17, 0.22) | -1.53%, -0.014 (-0.019, -0.01) * | 73.9%, 0.36 (0.33, 0.41) | 3.06%, 0 (0.00094, 0.01) * |

## Forecast for the 2022–2023 respiratory virus season

Figs 8 and 9 present real-time forecasts of October 2022 –March 2023 for the 10 states, and Table G in S1 Text shows a preliminary accuracy assessment based on data obtained on March 31, 2023. Accounting for under-detection, large numbers of infections (i.e., including undocumented asymptomatic or mild infections) were predicted in the coming months for most

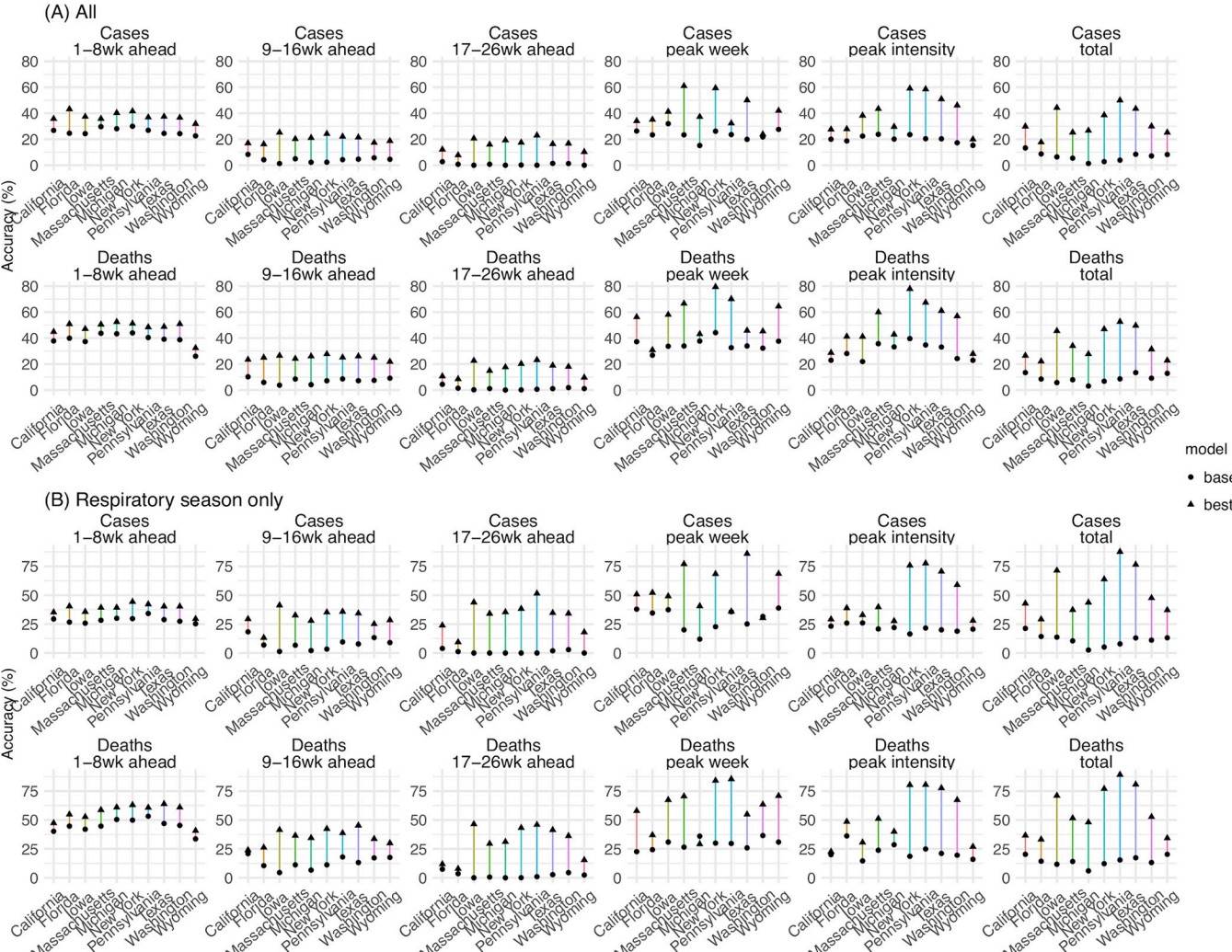

**Fig 7. Point prediction accuracy of the best-performing and baseline forecast systems.** Points show the average accuracy over all forecast weeks (A) or respiratory virus season (B). Results are aggregated by location (x-axis) and forecast target (panel subtitles) for cases (1st row) and deaths (2nd row, see panel subtitles) separately. Filled dots show the mean accuracy of forecasts generated using the baseline system; filled triangles show the accuracy of forecasts generated using the best-performing forecast system. The lines linking the two accuracies show the changes (mostly increases, as the triangles are more often above the dots), due to the combined application of the three proposed strategies (deflation, new variants, and transformed seasonality settings). Note all forecasts were generated retrospectively; to enable comparison of the model settings, mobility and vaccination data and estimates of infection detection rate and infection fatality risk during the forecast period were used (see main text for detail).

states; predicted attack rates over the 6-month prediction period ranged from 16% (IQR: 7–31%) in Florida to 30% (IQR: 15–47%) in Massachusetts (Fig 9). Relatively low case numbers and fewer deaths at levels similar to or lower than previous waves were forecast, assuming case ascertainment rates and infection-fatality risks similar to preceding weeks (Fig 9). Compared to data reported 6 months later (i.e., not used in the forecasts), the weekly forecasts in general captured trajectories of reported weekly cases over the 6 months for all 10 states (Fig 8, middle column for each state) but under-predicted deaths for half of the states (i.e., New York, Massachusetts, Michigan, Wyoming, and Florida; Fig 8, right column for each state). For the cumulative totals, predicted IQRs covered reported tallies in all 10 states for cases and the majority of states for deaths, while the 95% predicted intervals covered reported cumulative cases and deaths in all states (Fig 9).

**Table 2. Comparison of point prediction accuracy by the best-performing and the baseline forecast approaches.** Numbers show the mean point prediction accuracy of forecasts generated using the baseline v. the best-performing forecast approach; asterisk (*) indicates if the median of pairwise accuracy difference is significantly >0 or <0 at the α = 0.05 level, per a Wilcoxon rank sum test. Note all forecasts were generated retrospectively; to enable comparison of forecast approaches, mobility and vaccination data and estimates of infection detection rate and infection fatality risk during the forecast period were used (see main text for detail).

| State | Measure | 1-8wk ahead | 9-16wk ahead | 17-26wk ahead | peak week | peak intensity | total |
|---|---|---|---|---|---|---|---|
| All | Cases | 26% v 38% * | 4% v 20% * | 1% v 16% * | 24% v 42% * | 20% v 40% * | 7% v 33% * |
| All | Deaths | 39% v 48% * | 7% v 25% * | 1% v 16% * | 35% v 56% * | 30% v 50% * | 9% v 36% * |
| California | Cases | 27% v 36% * | 8% v 17% * | 3% v 12% * | 26% v 34% | 20% v 28% * | 13% v 30% * |
| California | Deaths | 38% v 45% * | 10% v 23% * | 4% v 11% * | 37% v 56% * | 23% v 29% | 13% v 26% * |
| Florida | Cases | 25% v 43% * | 4% v 16% * | 1% v 8% * | 23% v 35% * | 19% v 28% * | 9% v 18% * |
| Florida | Deaths | 40% v 51% * | 6% v 25% * | 1% v 8% * | 27% v 31% | 28% v 41% * | 8% v 22% * |
| Iowa | Cases | 24% v 37% * | 1% v 25% * | 0% v 21% * | 32% v 41% * | 22% v 38% * | 6% v 44% * |
| Iowa | Deaths | 37% v 47% * | 4% v 26% * | 0% v 23% * | 34% v 58% * | 22% v 41% * | 6% v 46% * |
| Massachusetts | Cases | 30% v 36% * | 5% v 20% * | 1% v 16% * | 23% v 61% * | 24% v 43% * | 5% v 25% * |
| Massachusetts | Deaths | 44% v 50% * | 8% v 24% * | 1% v 15% * | 34% v 67% * | 36% v 60% * | 8% v 34% * |
| Michigan | Cases | 28% v 40% * | 2% v 21% * | 0% v 19% * | 15% v 37% * | 20% v 30% * | 1% v 27% * |
| Michigan | Deaths | 43% v 52% * | 4% v 26% * | 0% v 18% * | 38% v 43% | 33% v 43% * | 3% v 28% * |
| New York | Cases | 30% v 42% * | 2% v 24% * | 0% v 18% * | 26% v 59% * | 24% v 59% * | 3% v 39% * |
| New York | Deaths | 44% v 51% * | 7% v 28% * | 0% v 20% * | 44% v 79% * | 40% v 78% * | 7% v 47% * |
| Pennsylvania | Cases | 27% v 37% * | 4% v 22% * | 0% v 23% * | 24% v 32% * | 20% v 59% * | 4% v 50% * |
| Pennsylvania | Deaths | 40% v 48% * | 9% v 25% * | 1% v 23% * | 33% v 70% * | 35% v 67% * | 9% v 53% * |
| Texas | Cases | 24% v 38% * | 5% v 22% * | 1% v 16% * | 20% v 50% * | 20% v 51% * | 8% v 44% * |
| Texas | Deaths | 39% v 49% * | 7% v 26% * | 1% v 19% * | 34% v 46% * | 33% v 61% * | 13% v 50% * |
| Washington | Cases | 24% v 37% * | 6% v 17% * | 1% v 17% * | 22% v 24% | 17% v 46% * | 7% v 30% * |
| Washington | Deaths | 39% v 51% * | 7% v 25% * | 2% v 18% * | 32% v 45% * | 24% v 57% * | 9% v 31% * |
| Wyoming | Cases | 23% v 32% * | 5% v 19% * | 0% v 10% * | 28% v 42% * | 15% v 20% * | 8% v 25% * |
| Wyoming | Deaths | 26% v 32% * | 9% v 22% * | 1% v 10% * | 38% v 64% * | 23% v 28% * | 13% v 23% * |

## Discussion

Given the uncertainties surrounding future SARS-CoV-2 transmission dynamics, it is immensely challenging to accurately predict long-lead COVID-19 epidemic outcomes. Here, we have proposed three strategies for sensibly improving long-lead COVID-19 forecast. Retrospective forecast accuracy is substantially improved using the three strategies in combination during both the pre-Omicron and Omicron periods, including for long-lead targets 6 months in the future. This improvement is consistent among 10 representative states across the US, indicating the robustness of the forecast method.

Our first strategy addresses the accumulation of forecast error over time. The simple deflation method proposed here substantially improves forecast accuracy across different model settings (here, different new variant and seasonality forms), time periods (pre-Omicron vs Omicron), and locations (different states). This consistent improvement indicates that deflation is effective in constraining outlier ensemble trajectories. Albeit possibly a severer issue for SARS-CoV-2 due to larger uncertainties and less constrained parameter estimates, error growth is a common challenge in forecasts of infectious diseases not limited to COVID-19 [21,22]. Future work could examine the utility of deflation in improving forecast accuracy for other infectious diseases.

Another challenge facing COVID-19 forecast derives from the uncertainty associated with new variant emergence. Based on past epidemiological dynamics of and population response to SARS-CoV-2 VOCs, we proposed a simple set of heuristic rules and applied them universally across time periods and locations. Despite their simplicity, the results here show that

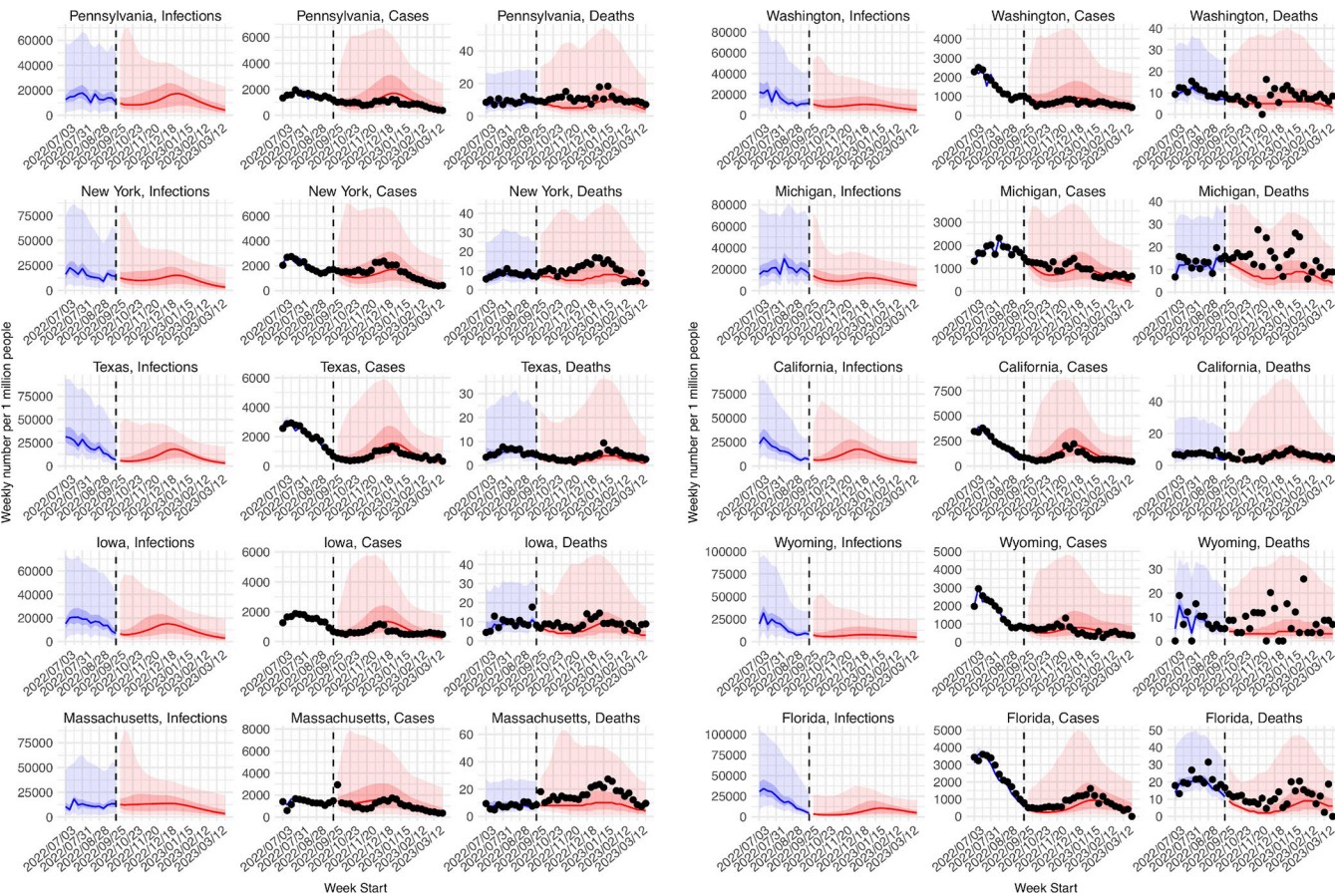

**Fig 8. Real-time forecasts for the 2022–2023 respiratory virus season.** The states are arranged based on accuracy of historical forecast (higher accuracy for those in the left panel and those on the top). In each panel, each row shows estimates and forecasts of weekly numbers of infections (1st column), cases (2nd column), or deaths (3rd column) for each state. Vertical dashed lines indicate the week of forecast initiation (i.e., October 2, 2022). Dots show reported weekly cases or deaths, including for the forecast period. Blue lines and blue areas (line = median; darker blue = 50% CI; lighter blue = 95% CI) show model training estimates. Red lines and red areas (line = median; dark red = 50% Predictive Interval; lighter red = 95% Predictive Interval) show model forecasts using the best-performing approach.

these heuristics substantially improved forecast accuracy compared to forecasts generated without them. These findings suggest that, while it is challenging to forecast the emergence of specific variants, the timing of future variant emergence and the impacts on key epidemiological characteristics (i.e., population susceptibility and virus transmissibility) can be learned from past VOC waves and used to support more accurate forecast. Much uncertainty remains regarding future SARS-CoV-2 genomic evolution and population immunity; however, the heuristics proposed here represent a first step anticipating the dynamic interplay of SARS-CoV-2 new variants and population immunity. Continued work to test the robustness of these heuristics as SARS-CoV-2 and population immunity continue to co-evolve is thus warranted.

The third focus of this study is seasonality. Several prior studies have examined the potential seasonality of COVID-19, using methods such as time series analyses, regression models, and sinusoidal functions [23–25]. However, the underlying mechanisms and likely nonlinear response to seasonal drivers are not fully characterized. Several concurrent changes including case ascertainment rate, NPIs and voluntary behavioral changes, and new variants further complicate such characterization. Here, we used local weather data along with a mechanistic model previously developed for influenza [16,17] to capture the nonlinear response of

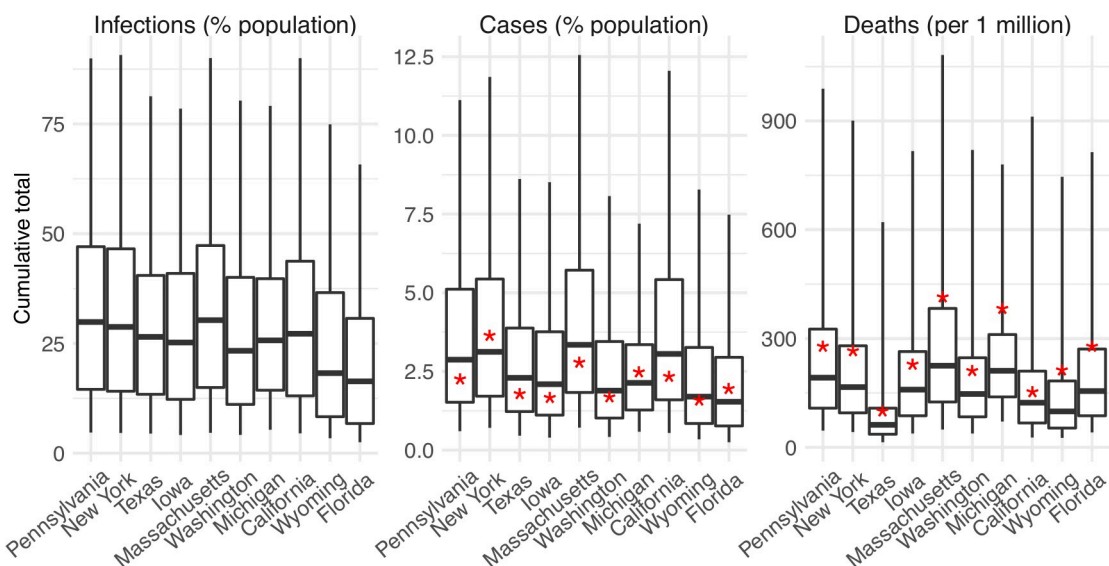

**Fig 9. Real-time forecasts of cumulative infections, cases, and deaths during the 2022–2023 respiratory virus season.** Box plots show distributions of predicted total number of infections (1st panel, scaled to population size; i.e. attack rate), cases (2nd panel, scaled to population size), and deaths (3rd panel, scaled per 1 million persons) from the week starting 10/2/2022 to the week starting 3/26/2023. Thick line = median; box edge = interquartile range; whisker = 95% prediction interval. The states (x-axis label) are arranged according to accuracy of historical forecast (higher accuracy from left to right). Red asterisks (*) show reported cumulative cases and deaths during the forecast period.

respiratory virus survival/transmission to humidity and temperature. In addition, we tested an alternative seasonality form given the likely differences between SARS-CoV-2 and influenza (e.g., likely higher infection risk for SARS-CoV-2 than influenza during the summer). When incorporated in a model-inference framework and forecast system, forecasts with both seasonality forms outperformed their counterpart without seasonality. Importantly, improvements during the respiratory season were consistent throughout the pandemic, including for the VOC waves, as well as for all 10 states with diverse climate conditions (Table D in S1 Text and S1 Fig). These findings indicate the robustness of the seasonality functions and the importance of incorporating seasonality in COVID-19 forecast. More fundamentally, the results support the idea that a common set of seasonal environmental/climate conditions influence the transmission dynamics of respiratory viruses, including but not limited to SARS-CoV-2 and influenza. Among the 10 states tested here, the transformed seasonality function tended to outperform the fixed seasonality function based on parameters estimated for influenza, except for Massachusetts and Michigan (Table D in S1 Text). This difference in performance suggests there are likely nuances in the seasonality of different respiratory viruses despite shared general characteristics.

To focus on the above three challenges, in our retrospective forecasts, we used data/estimates to account for several other factors shaping COVID-19 dynamics. These included behavioral changes (including those due to NPIs), vaccination uptake, changing detection rates and hence case ascertainment rate, as well as changes in infection fatality risk due to improvement of treatment, vaccination, prior infection, and differences in the innate virulence of circulating variants. For real-time forecast, such data and estimates would likely not be available and thus forecast accuracy would likely be degraded. Nonetheless, as societies emerge from the acute pandemic phase, many of these factors would likely reach certain norms (e.g., a relative stable fraction of the population may continue to adopt preventive measures and detection rates may stay low), reducing these uncertainties. Thus, though demonstrated mainly

retrospectively, the superior skill of the forecast methods developed here demonstrate means for generating more accurate and sensible long-lead COVID-19 forecasts.

## Methods

### Data used for model calibration

We used COVID-19 case and mortality data [26]–adjusted for circulating variants [27,28]–to capture transmission dynamics, mobility data [29] to represent concurrent NPIs, and vaccination data [30,31] to account for changes in population susceptibility due to vaccination.

For models including seasonality, we used weather data (i.e., temperature and humidity) [32,33] to estimate the infection seasonality trends. See detailed data sources and processing in S1 Text.

### Model calibration before forecast generation (i.e. inference)

The model-inference system is similar to systems we developed to estimate changes in transmissibility and immune erosion for SARS-CoV-2 VOCs including Alpha, Beta, Gamma, Delta, and Omicron [17–19]. However, to account for the fast waning of vaccine protection against infection and differential vaccine effectiveness (VE) against different variants, here we additionally accounted for variant-specific VE and waning vaccine protection against infection per Eq 1:

$$
\begin{cases}
\dfrac{dS}{dt} = \dfrac{R}{L_t} - \dfrac{b_t e_t m_t \beta_t IS}{N} - \varepsilon + \sum_{\tau=0}^{\tau=T} \rho_\tau V_{t-\tau} - \sum_{k=1}^{k=K} v_{k,t} \\[2mm]
\dfrac{dE}{dt} = \dfrac{b_t e_t m_t \beta_t IS}{N} - \dfrac{E}{Z_t} + \varepsilon \\[2mm]
\dfrac{dI}{dt} = \dfrac{E}{Z_t} - \dfrac{I}{D_t} \\[2mm]
\dfrac{dR}{dt} = \dfrac{I}{D_t} - \dfrac{R}{L_t} \\[2mm]
\dfrac{dV}{dt} = \sum_{k=1}^{k=K} v_{k,t} - \sum_{\tau=0}^{\tau=T} \rho_\tau V_{t-\tau}
\end{cases}
\tag{1}
$$

where $S$, $E$, $I$, $R$ are the number of susceptible, exposed (but not yet infectious), infectious, and recovered/deceased individuals; $N$ is the population size; and $\varepsilon$ is the number of travel-imported infections. To account for changes due to circulating variants, Eq 1 includes a time-varying transmission rate $\beta_t$, latency period $Z_t$, infectious period $D_t$, and immunity period $L_t$. To account for the impact of NPIs, Eq 1 uses the relative population mobility ($m_t$) to adjust the transmission rate and a scaling factor ($e_t$) to account for potential changes in effectiveness. To account for vaccination and waning, $V$ is the number of individuals vaccinated and protected from infection, $v_{k,t}$ is the number of individuals immunized after the $k$-th dose at time $t$ and $\rho_\tau$ is the probability of losing vaccine protection $\tau$ days post vaccination (see Supplemental Methods and Table E, both in S1 Text). As described below, $b_t$ is the seasonal infection risk at time $t$, depending on the seasonality setting. We further computed the number of cases and deaths each week to match with the observations using the model-simulated number of infections occurring each day (see S1 Text).

We ran the model jointly with the ensemble adjustment Kalman filter (EAKF [34]) and weekly COVID-19 case and mortality data to estimate the model state variables (e.g., $S$, $E$, and $I$) and parameters (e.g., $\beta_t$, $Z_t$, $D_t$, $L_t$, $e_t$). Briefly, the EAKF uses an ensemble of model realizations ($n$ = 500 here), each with initial parameters and variables randomly drawn from a *prior*

range (Table E in S1 Text). After model initialization, the system integrates the model ensemble forward in time for a week (per Eq 1) to compute the prior distribution for each model state variable, as well as the model-simulated number of cases and deaths for that week. The system then combines the prior estimates with the observed case and death data for the same week to compute the posterior per Bayes' theorem [34]. In addition, as in [17–19], during the filtering process, we applied space-reprobing [35], i.e., random replacement of parameter values for a small fraction of the model ensemble, to explore a wider range of parameter possibilities (Table E in S1 Text). The space-reprobing algorithm, along with the EAKF, allows the system to capture potential changes over time (e.g., increased detection for variants causing more severe disease, or increases in population susceptibility and transmission rate due to a new variant).

## Variations in forecast systems (deflation, new variant, and seasonality settings)

In total, 12 forecast approaches were tested (3 deflation levels × 2 new variants settings × 3 seasonality forms). The deflation algorithm is patterned after covariance inflation, as used in filtering methods [9,10]. However, unlike inflation applied during filtering (i.e., the model training period via data assimilation), deflation is applied during the forecast period. As the state variables change dynamically per the epidemic model (i.e., Eq 1), the error of some epidemic trajectories can amplify exponentially over time. Thus, here we applied deflation only to the state variables (i.e., not to the model parameters), per:

$$x_i^{def} = \gamma(x_i - \bar{x}) + \bar{x}, i = 1, \ldots, n \tag{2}$$

where $x_i$ is $i$-th ensemble member of a given state variable (here, S, E, or I) at each time step during the forecast period, before the deflation; $x_i^{def}$ is the corresponding "deflated" value; $\gamma$ is the deflation factor; and $\bar{x}$ is the ensemble mean of the state variable. Per Eq 2, deflation retains the ensemble mean, while reducing the ensemble spread to constrain error accumulation. In this study, we tested three levels of deflation, by setting $\gamma$ to 1 (i.e., no deflation), 0.95, and 0.9, separately.

To anticipate and account for potential surges and their impact on COVID-19 epidemic outcomes, we tested two approaches. The first, baseline approach simply assumes there are no changes in the circulating variant during the forecast period. For this approach, the forecasts were generated using the latest population susceptibility and transmission parameter estimates at the point of forecast initiation. For the second, new variant approach, we devised a set of heuristics to anticipate the likely timing and impact of new variant emergence during the forecast period, as detailed in the SI.

Seasonality is incorporated in the epidemic model (Eq 1) and applied throughout the model calibration (i.e., inference) and forecast periods. Here, we tested three seasonality settings. The first assumes no changes in seasonal risk of infection, by setting $b_t$ in Eq 1 to 1 for all weeks (referred to as "no seasonality"). The second seasonality form (termed "fixed seasonality") estimates the relative seasonality trend ($b_t$, same as in Eq 1) using local humidity and temperature data, based on the dependency of respiratory virus survival, including that of SARS-CoV-2, to temperature and humidity [12,17,18,36]; see Eq 3 and details, both in S1 Text. The third seasonality setting (termed "transformed seasonality") transforms the $b_t$ estimates to allow flexibility in the seasonal trend, including the peak timing, the number of weeks during a year with elevated infection risk, and the lowest risk level (see Eq 4 and details, both in S1 Text). Due to the lack of SARS-CoV-2 data to inform the parameter estimates, here we opted

to optimize the range for each parameter and used the best parameter ranges (see Supplemental Methods in S1 Text and S4 Fig) in the transformed seasonality model in the main analysis.

## Retrospective forecast

We tested the above 12 model-inference and forecast approaches (3 deflation levels × 2 new variants settings × 3 seasonality forms) for 10 states, i.e., California, Florida, Iowa, Massachusetts, Michigan, New York, Pennsylvania, Texas, Washington, and Wyoming. The 10 states span the 10 Health and Human Services (HHS) regions across the US, representing a wide range of population characteristics and COVID-19 pandemic dynamics (Fig 1). For all states, we generated retrospective forecasts of weekly cases and deaths 26 weeks (i.e., 6 months) into the future for the non-Omicron period and the Omicron period, separately. For the non-Omicron period, we initiated forecasts each week from the week of July 5, 2020 (i.e., after the initial wave) through the week of August 15, 2021. Note that because each forecast spans 6 months, the last forecasts initiated in mid-August 2021 extend to mid-Feb 2022, covering the entire Delta wave (see S1 Text for details). For the Omicron period, we initiated forecasts starting 5 weeks after local detection of Omicron BA.1 (roughly in early December 2021, depending on local data) through the week of September 25, 2022 (i.e., the last week of this study).

To generate a forecast, we ran the model-inference system until the week of forecast, halted the inference, and used the population susceptibility and transmissibility of the circulating variant estimated at that time to predict cases and deaths for the following 26 weeks (i.e., 6 months). Because the infection detection rate and infection-fatality risk are linked to observations of cases and deaths (see S1 Text), changes of these quantities during the forecast period could obscure the underlying infection rate and forecast accuracy. Thus, for these two parameters specifically, we used available model-inference estimates for corresponding forecast period weeks to allow comparison of model-forecast cases and deaths with the data while focusing on testing the accuracy of different model settings (e.g., seasonality and new variant settings). For the same reason, we used all available mobility and vaccination data including those for the forecast period, which would not be available in real time. For weeks in the future without data/estimates, we used the latest estimates instead. To account for model and filter stochasticity, we repeated each forecast 10 times, each time with initial parameters and state variables randomly drawn from the same prior ranges.

To evaluate forecast performance, we computed both the log score based on the probabilistic forecast and the accuracy of point prediction for 1) 1- to 26-week ahead prediction, and 2) peak week, 3) peak intensity, and 4) cumulative total over the 26-week forecast period, for cases and deaths, separately. Details are provided in S1 Text. Here, in brief, to compute the log score, we first binned the forecast ensemble to generate the forecast probability distribution Pr(x), and took the logarithm of the sum of Pr(x) across all related bins including the one including the observation ($bin^*$) and two adjacent ones ($bin^{*-1}$ and $bin^{*+1}$):

$$\log score = \log[\Pr(x)_{x \in bin^*} + \Pr(x)_{x \in bin^{*-1}} + \Pr(x)_{x \in bin^{*+1}}] \tag{5}$$

For the accuracy of point prediction, we deemed a forecast accurate (assigned a value of 1) if the median of the forecast ensemble is within ±1 week of the observed peak week or within ±25% of the observed case/death count, and inaccurate (assigned a value of 0) otherwise. As noted above, when aggregated over multiple forecasts, the average would represent the percentage of time a point prediction is accurate within these tolerances.

We compared the performance of each forecast approach (i.e., each of the 12 combinations of deflation, new variant setting, and seasonality form) overall or by forecast target, segregated by time period or respiratory virus season. To compute the overall score for each stratum, we

took the arithmetic mean of the log score or point prediction accuracy of forecasts generated by each forecast system, either across all forecast targets or for each target, over 1) all forecast weeks during the entire study period (i.e., July 2020 –September 2022), 2) the pre-Omicron period and Omicron period, separately, 3) the respiratory virus season (mid-October to mid-April, 6 months) and off season (the remaining 6 months), separately.

For pairwise comparison of forecast approaches, we computed the difference of log score or accuracy by simple subtraction of the two arithmetic-means. Relative difference was also computed. For the log score, the percent relative difference was computed as:

$$
\begin{aligned}
&\%relative\ difference\ in\ \log\ score \\
&= \frac{\exp\left(mean\ \log\ score\ of\ system\ a\right) - \exp\left(mean\ \log\ score\ of\ system\ b\right)}{\exp\left(mean\ \log\ score\ of\ system\ b\right)} \times 100\%
\end{aligned} \quad (6)
$$

As noted in [37], the exponent of the mean log score (Eq 5) can be interpreted as the probability correctly assigned to the bins containing the observations; thus Eq 6 gives the relative difference in the correctly assigned forecast probability. The percent relative difference in accuracy of point prediction was computed as:

$$
\begin{aligned}
&\%relative\ difference\ in\ accuracy \\
&= \frac{mean\ accuracy\ of\ system\ a - mean\ accuracy\ of\ system\ b}{mean\ accuracy\ of\ system\ b} \times 100\%
\end{aligned} \quad (7)
$$

In addition, we also computed the pair-wised difference of log score or accuracy (paired by forecast week for each target and location) and used boxplots to examine the distributions (see, e.g., Fig 6). We used the Wilcoxon rank sum test, a non-parametric statistical method, to test whether there is a difference in the median of the pair-wised differences [38].

## ARIMAX model forecast for comparison with approaches developed in this study

Auto-Regressive Integrated Moving Average (ARIMA) models and ARIMAX models (X represents external predictors) are commonly used to forecast different outcomes. For comparison with the approaches developed here, we also tested five ARIMA(X) models and used them to generate retrospective forecasts per the same procedure described above. The first model (i.e., simple ARIMA model) used weekly case or mortality data alone for model training. The remaining four models were ARIMAX models: 1) using case/mortality data and mobility data including for the forecast period (i.e., X = mobility; referred to as "ARIMAX.MOB"); 2) using case/mortality data and the estimated seasonal trend from the fixed seasonal model (i.e., X = seasonality; referred to as "ARIMAX.SN"); 3) using case/mortality data, mobility data, and the estimated seasonal trend (i.e., X = mobility and seasonality; referred to as "ARIMAX.MS"); and 4) using case/mortality data, mobility data, the estimated seasonal trend, and vaccination data (i.e., X = mobility, seasonality, and vaccination; referred to as "ARIMAX.FULL"). For vaccination, to account for the impact accumulated over time, we used cumulative vaccinations (here, in the past 3 months for cases, and the past 9 months for deaths). For model optimization, we used the "auto.arima" function of the "forecast" R package [39], which searches all possible models within the specified order constraints (here, we used the default settings) to identify the best ARIMA(X) model (here, based on the corrected Akaike information criterion by default).

For the five model forms, the ARIMAX.FULL model (i.e., including mobility, seasonality, and vaccination) was only able to generate forecasts for less than half of the study weeks as the

auto.arima function was unable to identify parameters for this model. The other four models were able to generate forecasts for most study weeks and across the entire study period, the ARIMAX.SN (i.e., seasonality included) performed the best (see Table E in S1 Text). As such, we used the ARIMX.SN model as a benchmark model for comparison with approaches developed in this study.

## Preliminary assessment of real-time forecasts for the 2022–2023 respiratory virus season

The last forecasts in this study were generated using all available data up to the week starting October 2, 2022 and spanned 6 months through the week starting March 26, 2023. These were real-time forecasts generated without future information. We assess these real-time forecasts using data downloaded on March 31, 2023 (1 day after the data release). Since these data may be revised in the future (*n.b.* data revision after the initial release has been common), we consider the assessment preliminary. As detailed in S1 Text, we used case and mortality data from the New York Times (NYT; [26]) for model calibration prior to generating the forecasts. However, in the six months since the initial study, NYT data have become more irregular for some states, likely due to infrequent data reporting and updating. As such, for this preliminary assessment, we instead used data from the Centers for Disease Control and Prevention (CDC) [40], except for mortality in Washington State for which the CDC data appeared to be mis-dated whereas NYT data and mortality data from the Center for Systems Science and Engineering (CSSE) at Johns Hopkins University [41] were consistent with each other. In addition, the CDC data were aggregated for each week from Thursday to Wednesday, rather than Sunday to Saturday. To enable the comparison, we thus shifted the dates of the CDC data 3 days.

All inference, forecast, and statistical analyses were carried out using the R language (https://www.r-project.org).

## Supporting information

**S1 Text. Supplemental methods and tables.**
(DOCX)

**S1 Fig. Comparison of seasonality forms.** For each state (each panel), the blue line shows the estimated trend of seasonal infection risk using Eqs 3a-b and location weather data (temperature and humidity). Grey lines show 100 examples of the transformed seasonal trends per Eqs 4a-d with parameters randomly sampled from the best parameter ranges (S4 Fig); the black line shows the mean of the 100 example trends.
(TIF)

**S2 Fig. Impact of deflation on probabilistic forecast of different targets.** Heatmaps show differences in mean log score for cases (A) and deaths (B), between each forecast approach with different deflation settings (deflation factor $\gamma = 0.95$ vs none in the 1st row, 0.9 vs none in the 2nd row, and 0.9 vs 0.95 in the 3rd row; see panel subtitles). Results are aggregated over all forecast weeks for each type of target (y-axis), forecast approach (see specific settings of new variants and seasonality in subtitles), and location (x-axis). For each pairwise comparison (e.g., 0.95 vs none), a positive difference indicates the former approach (e.g., 0.95) outperforms the latter (e.g., none).
(TIF)

**S3 Fig. Impact of deflation on point estimate accuracy of different targets.** Heatmaps show differences in forecast accuracy of point estimates for cases (A) and deaths (B), between each

forecast approach with different deflation settings (deflation factor γ = 0.95 vs none in the 1$^{st}$ row, 0.9 vs none in the 2$^{nd}$ row, and 0.9 vs 0.95 in the 3$^{rd}$ row; see panel subtitles). Results are aggregated over all forecast weeks for each type of target (y-axis), forecast approach (see specific settings of new variants and seasonality in subtitles), and location (x-axis). For each pairwise comparison (e.g., 0.95 vs none), a positive difference indicates the former approach (e.g., 0.95) outperforms the latter (e.g., none).
(TIF)

**S4 Fig. Comparison of forecast performance using the transformed seasonality function, with different parameter ranges.** The parameter ranges are shown in x-axis labels for the three parameters in Eq 4a-d (from bottom to top: $p_{shift}$, $\delta$, and $b_{t,\,lwr}$). 'x's indicate the best parameter ranges for the corresponding state.
(TIF)

## Author Contributions

**Conceptualization:** Wan Yang.

**Formal analysis:** Wan Yang.

**Funding acquisition:** Wan Yang, Jeffrey Shaman.

**Investigation:** Wan Yang, Jeffrey Shaman.

**Methodology:** Wan Yang.

**Writing – original draft:** Wan Yang.

**Writing – review & editing:** Wan Yang, Jeffrey Shaman.

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
