## [Decision Letter · Decision Letter 0]

3 Mar 2023

Dear Dr. Yang,

Thank you very much for submitting your manuscript "Development of Accurate Long-lead COVID-19 Forecast" for consideration at PLOS Computational Biology.

As with all papers reviewed by the journal, your manuscript was reviewed by members of the editorial board and by several independent reviewers. In light of the reviews (below this email), we would like to invite the resubmission of a significantly-revised version that takes into account the reviewers' comments.

We cannot make any decision about publication until we have seen the revised manuscript and your response to the reviewers' comments. Your revised manuscript is also likely to be sent to reviewers for further evaluation.

Sincerely,

Benjamin Althouse

Academic Editor

PLOS Computational Biology

Virginia Pitzer

Section Editor

PLOS Computational Biology

Reviewer's Responses to Questions

**Comments to the Authors:**

Reviewer #1: In this paper, the authors aim to develop strategies to enhance forecasts of the trajectory of the COVID-19 pandemic. Specifically, an “optimized approach” is used to derive improved predictions 6 months ahead using data for 10 representative US states. My specific comments follow.

1. After reading the paper, I was unclear about the “optimization” framework employed to generate the prediction enhancements. For instance, authors explore three “deflation” factors (1.0, 0.95, 0.9). However, these values were chosen arbitrarily but resulted in positive improvements in forecasting performance relative to their baseline model. If an optimization framework is employed, the model should be calibrated with a training dataset based on the first part of the pandemic (e.g., 2020 and 2021) and then assess forecasting performance based on the most recent pandemic data (2022).

2. Regarding the forecasting horizon, epidemic models have struggled to generate reasonably accurate short-term (up to 4 weeks ahead) predictions of the COVID-19 pandemic. Some real-time and retrospective efforts on this are now well documented as the authors are aware (CDC Forecasting hub), and their forecasting results are publicly available. Hence, it is essential to compare the forecasting performance of any new models/approaches, such as the one presented here, with the historical performance of prior modeling efforts. This is to say that before assessing such long-lead predictions, it would be important to evaluate short-term predictions (e.g., 4-week) and compare the results with those obtained from prior forecasting efforts to gauge the extent of the forecasting performance improvements reported in this paper. Without comparisons with a benchmark model is challenging to assess the importance of the advancement reported using new models or approaches.

3. It’d be helpful to to compare their forecasting performance results with a benchmark model such as ARIMA. To what extent the model improves performance relative to more straightforward models? It will help determine how successful the approach is comparable to other models, even if retrospective, which is critical to advancing the field of epidemic forecasting.

4. A log score is used as the primary metric to assess forecasting performance. However, the gold standard to determine forecasting performance is the weighted interval score, which is a proper score and has been used to evaluate performance in prior forecasting efforts during the COVID-19 pandemic.

5. Reporting the performance metrics in a supp file will help other teams use your model as a benchmark for comparison purposes.

6. The description of the epidemic model could be enhanced by providing a table that indicates which parameters are estimated and which are fixed from external information.

7. The paper's title should indicate that the forecasts are based on a retrospective analysis.

Reviewer #2: The authors present a very nice and comprehensive study exploring model improvements to build a better COVID-19 forecasting model. The paper is very well written and highly detailed, yet interesting and readable. While I do not expect any major revisions, there are a couple concerns that I would like to see addressed.

First, the authors made the choice to use data available and even fitted separately from the "future" forecast period. While I understand the choice to do this in their analysis aimed to understand how each of the 3 components they were evaluating contributed to forecast accuracy, I would like to see either a little more text discussing this, or even better, a sensitivity analysis where those future data are not used, and instead they are either predicted or used from the point of forecast. While it may not make a major difference, these predictions may interact with the impact of each of these components. Further, for readers who may not read in depth, they may misinterpret the model as being much more accurate than it would be to forecast a long horizon. This should be made more explicit.

Second, it is not clear to me why specific periods of the pandemic are not included in the analysis, in particular from August - November 2021. While this period may have been challenging due to the rise in Omicron, it seems a little unfair to not present a period because the model did not perform well during it, if that is the case. Either a reason for exclusion should be made clear, or it should be included.

**Have the authors made all data and (if applicable) computational code underlying the findings in their manuscript fully available?**

Reviewer #1: Yes

Reviewer #2: Yes

PLOS authors have the option to publish the peer review history of their article (what does this mean?). If published, this will include your full peer review and any attached files.

Reviewer #1: No

Reviewer #2: No
---

## [Decision Letter · Decision Letter 1]

2 Jun 2023

Dear Dr. Yang,

Thank you very much for submitting your manuscript "Development of Accurate Long-lead COVID-19 Forecast" for consideration at PLOS Computational Biology. As with all papers reviewed by the journal, your manuscript was reviewed by members of the editorial board and by several independent reviewers. The reviewers appreciated the attention to an important topic. Based on the reviews, we are likely to accept this manuscript for publication, providing that you modify the manuscript according to the review recommendations.

One reviewer had a couple items for your consideration. Once these have been addressed, we should be able to accept the manuscript without further review.

Sincerely,

Benjamin Althouse

Academic Editor

PLOS Computational Biology

Virginia Pitzer

Section Editor

PLOS Computational Biology

Reviewer's Responses to Questions

**Comments to the Authors:**

Reviewer #1: Thanks for the clarifications, the extension of the study to incorporate real-time forecasts, and the additional benchmark comparison with the ARIMA forecasts, which I believe is an essential addition to the paper indicating that the method presented outperformed ARIMA forecasts.

I only have a few residual comments for the authors’ consideration.

1)While the scope of the authors’ work is on long lead forecasts, I did not entirely agree that it would be unfair to retrospectively compare the performance of new models/approaches with the results of prior real-time studies, such as the CDC COVID-19 forecasting hub so long the new methods are fully documented/reproducible. The findings should also be presented as retrospective rather than derived in real-time.

2)Regarding the performance metrics, as noted previously more recent forecasting studies including the CDC COVID-19 forecasting hub have based the performance evaluation on the weighted interval score (WIS) while previous forecasting studies, including those noted by the authors, used to focus performance on the log score as indicated by the authors. The authors may agree that using a standard set of performance metrics would allow authors to systematically compare performance across different methods/approaches that may be developed subsequently.

**Have the authors made all data and (if applicable) computational code underlying the findings in their manuscript fully available?**

Reviewer #1: Yes

PLOS authors have the option to publish the peer review history of their article (what does this mean?). If published, this will include your full peer review and any attached files.

Reviewer #1: No

Figure Files:

Data Requirements:

Reproducibility:

References:

---

## [Editor Report · Decision Letter 2]

16 Jun 2023

Dear Dr. Yang,

We are pleased to inform you that your manuscript 'Development of Accurate Long-lead COVID-19 Forecast' has been provisionally accepted for publication in PLOS Computational Biology.

Best regards,

Benjamin Althouse

Academic Editor

PLOS Computational Biology

Virginia Pitzer

Section Editor

PLOS Computational Biology

---

## [Editor Report · Acceptance letter]

5 Jul 2023

PCOMPBIOL-D-22-01834R2 

Development of Accurate Long-lead COVID-19 Forecast

Dear Dr Yang,

I am pleased to inform you that your manuscript has been formally accepted for publication in PLOS Computational Biology. Your manuscript is now with our production department and you will be notified of the publication date in due course.

With kind regards,

Zsofi Zombor
